# OpenReview forum: "Adaptive Residual-Update Steering for Low-Overhead Hallucination Mitigation in Large Vision-Language Models"
_ICML.cc/2026/Conference — ICML 2026 regular_

### Official Review · Reviewer_Jaaz · 2026-03-07

[review text omitted: it was posted to a different submission]

---

> ### Author Rebuttal · Authors · 2026-03-30
>
> ### **All rebuttal figures:** https://anonymous.4open.science/r/Rebuttal-Figures-for-ICML-Submission-19687-78CF/README.md
>
>
> We thank the reviewer for the positive evaluation and recognition of CARD's novelty and low overhead.
>
> ---
>
> **W1: Evidence that the visual signal is preserved during prefill**
>
> > The paper claims that the visual signal is maximally preserved in the residual stream during the prefill stage. However, there is no clear experimental evidence.
>
> We provide multiple lines of evidence:
>
> **(a) NEW — Semantic Clustering [Fig R1]:** We extracted CARD vectors for 4 COCO categories (airplane, elephant, pizza, train; 50 images each). Both t-SNE and PCA show clear clustering, demonstrating that CARD captures visual semantic content rather than noise.
>
> **(b) NEW — Residual Direction Alignment [Fig R2, center panel, n=40,137]:** We ask: does adding an image shift CARD in a consistent direction? We compare image-conditioned CARD to the shift direction ($v_{\text{CARD}}^{\text{img}} - v_{\text{CARD}}^{\text{text}}$). Result: cos = 0.548 ± 0.011. The tight std confirms images push the CARD toward a shared direction, not randomly.
>
> **(c) NEW — Rotation Angle [Fig R2, right panel]:** We then ask: how much does adding an image change CARD? The angle between text-only and image-conditioned CARD is 66.4° ± 1.5° on LLaVA-1.5-7B. Every image rotates CARD by almost the same amount (std = 1.5°), confirming a systematic visual signal.
>
> **(d) Existing — Appendix A.4**, summarized for convenience:
>
> - Fig 5a shows paired t-SNE overlay: each image's CARD systematically rotates away from its text-only counterpart, with lines connecting the same samples — a clear directional shift, not noise.
> - Fig 6a quantifies this rotation as ~40° on Idefics2 with tight distribution.
> - Fig 6b shows the image-conditioned CARD is closer to the actual steering direction than the text-only CARD (alignment gain = 0.239), confirming CARD captures the signal the Beta Gate relies on.
>
> **(e) Internal Dynamics** (Appendix B.1, Figure 7): Residual update magnitudes peak in late layers during prefill, consistent with active visual processing.
>
> **(f) Mechanistic:** During prefill, all image and text tokens are processed jointly before autoregressive dilution, making it by construction the point of maximal visual-textual fusion.
>
> ---
>
> **W2: Underdeveloped preliminary section**
>
> > The preliminary section does not provide enough introduction to key concepts and technical foundation.
>
> We will expand Section 3.1 with: (1) a clearer definition of the residual stream; (2) primer on prefill vs. decode; (3) connection to the visual dilution problem.
>
> ---
>
> **W3: Placement of Table 4**
>
> > Table 4 should be placed in a more prominent position.
>
> Agreed. We will move Table 4 to a more prominent position.
>
> ---
>
> **W4: Theoretical complexity**
>
> > A theoretical analysis of computational complexity should be included.
>
> CARD extraction requires $O(T_{\text{pre}} \times d)$ for pooling over $T_{\text{pre}}$ prefill tokens of dimension $d$, plus $O(d)$ for normalization. Both are computed during the mandatory prefill pass at zero extra cost.
>
> Per decode step, the Beta Gate requires $O(d)$: one dot product for cosine similarity, two softplus activations, and one scalar-vector multiply. Over a generated sequence of $T_{\text{dec}}$ tokens, this totals $O(T_{\text{dec}} \times d)$.
>
> For comparison, each transformer layer costs $O(T_{\text{dec}} \times d^2)$ for self-attention and FFN. The gate adds $O(T_{\text{dec}} \times d)$, giving a relative overhead of $\frac{O(d)}{O(d^2)} = O(1/d)$ per layer. With $d = 4096$ for a 7B model, this is negligible, consistent with the <4% latency in Table 4.

---

> > ### Author Rebuttal · Reviewer_Jaaz · 2026-04-01
> >
> > Thanks. Most of my concerns are well-solved except the preliminary section part, so i choose to keep my positive score.

---

> > > ### Author Response · Authors · 2026-04-02
> > >
> > > Thank you for confirming your positive assessment.
> > >
> > > We take the feedback on the preliminary section seriously and will substantially revise Section 3.1 in the camera-ready version. Specifically, we plan to:
> > >
> > > 1. **Residual stream definition:** We will formally define the residual stream and explain how each sublayer (self-attention, FFN) contributes an additive update.
> > >
> > > 2. **Prefill vs. decode distinction:** We will add a clear explanation of the two-stage autoregressive process: how prefill processes all image and prompt tokens in parallel, while decode generates tokens one at a time.
> > >
> > > 3. **Visual dilution problem:** We will formally motivate the visual dilution phenomenon with reference to prior work, explaining how the visual signal fades as the generated sequence grows, and why this motivates extracting a persistent visual anchor from the prefill stage.
> > >
> > > 4. **Notation table:** We will add a summary table of key notation ($v_{\text{CARD}}$, $g_t$, $\alpha_{\max}$, $s_t$, etc.) to make the method section easier to follow.
> > >
> > > We believe these additions will make the paper more self-contained and accessible to readers less familiar with the internal mechanics of transformer-based LVLMs.
> > >
> > > Thank you again for the constructive suggestion.

---

### Official Review · Reviewer_nowv · 2026-03-12

**Soundness:** 3
**Presentation:** 3
**Significance:** 2
**Originality:** 3
**Overall Recommendation:** 4
**Confidence:** 4

**Summary:**

This paper proposes Residual-Update Directed DEcoding Regulation (RUDDER), an efficient method for reducing hallucination by injecting a vector extracted from the prefill stage of the LVLM. The method also employs a gating mechanism to control the strength of the injected steering vector. The authors evaluate the effectiveness of their approach on standard hallucination benchmarks and further assess its efficiency, showing that the method can reduce hallucination while introducing low computational overhead.

**Compliance With Llm Reviewing Policy:**

Affirmed.

**Final Justification:**

The rebuttal clarified most issues and better motivated the method, especially the layer selection, pooling design, and hidden-state intervention. My remaining concern about the beta gate is reduced, though I still find this part less fully justified than the others, particularly in binary settings like POPE. Overall, the rebuttal was helpful, and I keep my score at weak accept.

**Key Questions For Authors:**

Why should simple average pooling over token-level residual updates remain valid across architectures with very different token structures and multimodal fusion designs?

How does injecting a normalized CARD vector into the residual stream affect the hidden-state scale, and could this introduce a mismatch with the model’s native residual updates?

**Limitations:**

yes

**Strengths And Weaknesses:**

Strenghts

1. The paper is well written, easy to follow, and the proposed method is clearly presented.

2. The experimental evaluation is comprehensive, as the authors assess the method on standard hallucination benchmarks and also show that performance on more general multimodal tasks is largely preserved.

3. The efficiency analysis is a strong point of the paper, since it shows that the method reduces hallucination while introducing low computational overhead.

Weaknesses:

1. The procedure for selecting the CARD layer on the validation set is not clearly explained. The paper states that the layers are swept to maximize alignment, but it is unclear what this alignment is computed with respect to, how it is measured, and why it is a reliable criterion for selecting the intervention layer. In addition, the selected layers differ substantially across models, for example between InstructBLIP and LLaVA. It is difficult to assess whether the same strategy would generalize to other LVLMs.

2. The use of average pooling to construct the CARD vector appears to be a fairly general heuristic, but different architectures can have very different token structures and multimodal fusion mechanisms. For example, Qwen-style models may already include mean pooling or different visual token behavior, whereas LLaVA may contain a larger number of visual tokens. In such cases, averaging token-level updates may correspond to very different representations across architectures. Therefore, it is not clear that the proposed pooling strategy is generally valid.

3. It is also not fully clear how the gating design helps reduce hallucination. The method increases steering when the hidden state is already aligned with the visual evidence, and suppresses intervention when the alignment is low. However, hallucinated tokens may be precisely those with low visual grounding. In such cases, the method appears to reduce steering at the point where correction may actually be needed. As a result, the mechanism seems more suited to reinforcing already grounded tokens than to actively correcting hallucinated ones.

---

> ### Author Rebuttal · Authors · 2026-03-30
>
> ### **All rebuttal figures:** https://anonymous.4open.science/r/Rebuttal-Figures-for-ICML-Submission-19687-78CF/README.md
>
>
> We thank the reviewer for the positive assessment and insightful questions.
>
> ---
>
> **W1:**
>
> > The procedure for selecting the CARD layer is not clearly explained. The selected layers differ substantially across models.
>
> The theoretical goal is to identify the layer in which residual updates most faithfully encode visual evidence. In practice, we validate hyperparameteres using 100 held-out MSCOCO images in three steps:
>
> **Step 1 (Coarse sweep):** Sweep every 3rd layer (e.g., L=1,4,7,...,31), extract CARD at each, apply steering with a deliberately strong fixed setting, and measure the performance in the validation set. This identifies the most effective region (e.g., L=28 on Idefics2).
>
> **Step 2 (Fine sweep):** Sweep adjacent layers around the coarse best (e.g., L=26,27,28,29,30), selecting the optimal layer (Figure 3).
>
> **Step 3 (Tuning):** With the layer fixed, grid-search ($α_{max}$, $k$) for optimal trade-off (Section 4.1).
>
> Since RUDDER operates within a single forward pass with <4% overhead, each configuration runs at nearly vanilla speed, making the entire calibration process lightweight (~100 images, completes in minutes).
>
> **Why layers differ across models:** InstructBLIP's Q-Former compresses visual info before the decoder → visual signal peaks at early layers (L=1). LLaVA's linear projection lets visual info propagate deeper → signal peaks at mid-to-late layers (L=30, consistent with Figure 7). General rule: early compressed fusion → early layers; late/linear fusion → later layers.
>
> ---
>
> **W2:**
>
> > Different architectures have very different token structures. Averaging token-level updates may correspond to very different representations.
>
> To clarify: (1) we pool residual updates ($\Delta_i = A_i^l$), not raw hidden states, and (2) we use $\|\Delta_i\|$-weighted pooling (Eq. 2), not simple averaging. The residual update isolates the net contribution of self-attention at each position, separating the visual influence signal from architecture-specific token structures and fusion mechanisms. The norm-weighting ensures that tokens with stronger visual influence dominate the aggregate, naturally adapting to each architecture's token behavior: in LLaVA (576 visual tokens with linear projection), many tokens carry a moderate signal; in InstructBLIP (32 Q-Former tokens), fewer tokens carry a concentrated signal. The weighting compensates for this automatically.
>
> Empirically, the consistent improvements across all four architectures in Tables 1-2 confirm that the strategy generalizes. Our new clustering experiment [Fig R1] provides further direct evidence: CARD vectors from 4 MSCOCO categories show clear semantic clustering, proving the pooled vectors capture meaningful visual content rather than noise.
>
> ---
>
> **W3:**
>
> > The method increases steering when alignment is high and suppresses when low. Hallucinated tokens may have low visual grounding, so the method appears to reduce steering where correction is needed.
>
> The Beta Gate is a preventive trust mechanism, not a reactive error corrector:
>
> **(1)** It prevents hallucination by continuously reinforcing the visual trajectory, not waiting for errors.
>
> **(2)** The gate tracks visual grounding strength, not just syntax. From a skiing image [Fig R3]:
>
> ```
> Token    |  g_t | cos  | Evidence
> skiing   | .836 | .287 | ★ visible
> slope    | .816 | .262 | ★ visible
> flags    | .809 | .258 | ★ visible
> control  | .664 | .122 | ✗ abstract
> high     | .656 | .115 | ✗ not in image
> man      | .652 | .112 | △ small/blurry
> ```
>
> The gate gives lower values to tokens lacking direct visual evidence — abstract concepts, hard-to-see objects — not just function words.
>
> **(3)** Gate floor ($g_{min}$=0.05) ensures steering is never fully suppressed.
>
> **(4)** RUDDER-Beta outperforms RUDDER-Add (fixed-strength) on CHAIR (Table 1). If suppressing low-alignment tokens missed hallucinations, fixed-strength would win. The opposite is observed.
>
> ---
>
> **Q1: We use norm-weighted pooling, not simple mean** Our default uses ||Δ_i||-weighted pooling(Eq. 2). Each token's residual update is weighted by its L2 norm, naturally prioritizing tokens with stronger visual influence.
>
> **Q2: Scale mismatch**
>
> > How does injecting a normalized CARD vector into the residual stream affect the hidden-state scale?
>
> Injection magnitude = $α_{max} × g_t$ (Eq. 4). With α_max=20 and gate values 0.65-0.84 (Experiment 3), the steering norm is ~13-17, within the native residual update range of ~12-18 at layer 30 (Figure 7a).
>
> Our adaptive gate provides implicit scale matching: low-alignment tokens receive weaker injection (down to $α_{max} × g _ {min} $ = 1.0). MME results (Table 3) confirm general capabilities preserved. Notably, competing methods (VISTA, ASD) use fixed-strength injection without scale adaptation.

---

> > ### Author Rebuttal · Reviewer_nowv · 2026-04-02
> >
> > Thank you to the authors for the detailed rebuttal.
> >
> > The rebuttal addressed most of my concerns, except for W3. I still do not fully understand the proposed prevention mechanism. In particular, in the POPE benchmark, the model’s output is only a single token, such as “yes” or “no”. If that token is visually grounded, the beta gate reinforces it; however, if it is not visually grounded, the gate reduces the intervention. In this case, it is unclear to me how the mechanism helps at the exact point where correction may be needed most.

---

> > > ### Author Response · Authors · 2026-04-02
> > >
> > > Thank you for the follow-up. This is an important question.
> > >
> > > You are correct that "yes" and "no" carry no visual content. However, RUDDER does not steer the output token — it steers the hidden states at layer $L$ during the forward pass, shaping the representation that determines whether to answer "yes" or "no."
> > >
> > > The CARD vector is extracted from prefill residual updates over both image and prompt tokens jointly (Eq. 2). On POPE, the prompt is "Is there a \<object\> in the image?", so CARD encodes the interaction between the image and the queried object. When the object is present, this interaction is strong → higher gate → stronger steering. When absent, the interaction is weak → lower gate → weaker steering. The gate modulates based on image-question alignment rather than on the output word.
> > >
> > > ---
> > >
> > > Regarding the concern that the gate reduces steering when correction is most needed: RUDDER's benefit on POPE is not about correcting individual wrong answers; it is about consistently reinforcing visual evidence when it is present, making the model, overall, more trusting of what it sees. The recall and F1 improvements across all 15 configurations confirm this.
> > >
> > > ---
> > >
> > > Our new POPE hyperparameter data sheds light on how RUDDER operates in this setting. Across all 15 configurations, RUDDER-Beta makes the model more willing to say "yes", as recall increases from 0.780 to 0.796–0.823, meaning the model correctly identifies more objects that are actually present. This comes with a small trade-off in precision (0.928 → 0.904–0.917), but the net effect is positive: accuracy and F1 improve across all configurations. In other words, by reinforcing visual evidence during the forward pass, RUDDER helps the model trust what it sees, reducing the tendency to default to "no" when uncertain.
> > >
> > > On LLaVA-1.5, RUDDER-Beta outperforms RUDDER-Add on POPE (Acc: 86.5 vs 85.9, F1: 86.0 vs 85.0, Table 2). On InstructBLIP, RUDDER-Add is slightly better, consistent with our analysis in Section 4.2.3: InstructBLIP's Q-Former already compresses visual information into a compact representation, making uniform steering sufficient for simple binary tasks. The adaptive gate provides greater benefit on architectures like LLaVA, where visual tokens are more numerous and diffuse.

---

### Official Review · Reviewer_jMxp · 2026-03-12

**Soundness:** 3
**Presentation:** 3
**Significance:** 2
**Originality:** 2
**Overall Recommendation:** 4
**Confidence:** 3

**Summary:**

In this paper, the authors address the problem of object hallucination in Large Vision-Language Models (LVLMs), which often arises due to the gradual dilution of visual information during autoregressive text generation. To mitigate this issue while maintaining high inference efficiency, the authors propose a lightweight inference-time intervention framework called Residual-Update Directed Decoding Regulation (RUDDER). The method extracts a Contextual Activation Residual Direction (CARD) vector from residual updates during the prefill stage, capturing a per-sample visual evidence direction that serves as a persistent visual anchor throughout decoding. During generation, an adaptive Beta Gate mechanism modulates the injection strength of the CARD vector based on the alignment between the current hidden state and the visual evidence, enabling token-wise adaptive steering toward visually grounded outputs. The proposed approach operates within a single forward pass and introduces negligible computational overhead. Extensive experiments on multiple LVLM architectures and benchmarks, including CHAIR and POPE, demonstrate that RUDDER effectively reduces hallucinations while preserving general multimodal capabilities and maintaining high inference throughput. Detailed comments are provided below.

**Compliance With Llm Reviewing Policy:**

Affirmed.

**Final Justification:**

The paper proposes a simple and efficient training-free inference-time method for mitigating hallucinations in LVLMs, addressing an important and practical problem. The approach is technically sound and demonstrates consistent improvements with minimal computational overhead. While the originality is somewhat limited and the empirical evaluation has some scope constraints, the overall contribution is still meaningful. The authors’ rebuttal effectively addressed my main concerns, particularly by providing additional empirical validation and analyses, which increased my confidence in the method. Therefore, I update my recommendation to 4 (Weak Accept).

**Key Questions For Authors:**

1.	Provide empirical validation that the extracted CARD vector aligns with the latent representation shift induced by visual inputs (e.g., the difference between image-conditioned and text-only hidden states).

2.	Demonstrate that the chosen hyperparameters remain close to optimal across different tasks to support the claim that the method is task-agnostic.

3.	Provide analyses or visualizations of token-level Beta Gate values during generation to illustrate how the gating mechanism behaves.

4.	Evaluate the proposed method on more recent LVLM architectures (e.g., newer Qwen-VL models) to demonstrate its generality.

5.	Include qualitative comparisons with additional hallucination mitigation baselines to better illustrate the advantages of the proposed method.

**Limitations:**

Yes.

**Strengths And Weaknesses:**

The paper proposes a training-free inference-time steering method to mitigate hallucinations in large vision-language models.

1.	The method introduces a simple and training-free approach that extracts a visual evidence direction from residual updates and injects it during decoding to reduce hallucinations.

2.	The approach is computationally efficient and can be integrated into existing LVLMs without retraining, while achieving consistent improvements on several hallucination benchmarks.

However, several aspects limit the strength of the empirical and methodological conclusions.

1.	The paper assumes that the pooled residual updates extracted during the prefill stage represent a reliable visual evidence direction. However, this claim lacks strong empirical validation. A more convincing analysis would compare the extracted CARD vector with the latent representation shift induced by visual inputs (e.g., the difference between image-conditioned and text-only hidden states) to verify whether the two directions are consistent.

2.	The hyperparameters are tuned using the CHAIR benchmark and then directly applied to other tasks such as POPE and MME. However, the paper does not verify whether this configuration is near-optimal for these tasks. Additional experiments demonstrating that the same hyperparameters remain effective across tasks would better support the claim that the method is task-agnostic.

3.	The evaluation mainly focuses on object hallucination benchmarks (e.g., CHAIR and POPE), while other types of multimodal errors, such as attribute hallucinations or relational reasoning mistakes, are not thoroughly evaluated.

4.	The experimental evaluation is mainly conducted on relatively earlier LVLM backbones (e.g., LLaVA-1.5, Idefics2, and InstructBLIP). It remains unclear whether the proposed method generalizes to more recent multimodal architectures such as the Qwen3-VL series.

5.	While the adaptive Beta Gate is designed to regulate token-wise intervention strength, the paper lacks detailed analysis or visualization illustrating how the gating mechanism behaves during generation (e.g., token-level gate values or alignment trends).

6.	The qualitative examples in the appendix only compare the vanilla model and RUDDER. Including additional hallucination mitigation baselines in the qualitative comparisons would provide a clearer and more comprehensive illustration of the advantages of the proposed method.

---

> ### Author Rebuttal · Authors · 2026-03-30
>
> We thank the reviewer for the detailed feedback. We conducted new experiments during the rebuttal period.
>
> **All rebuttal figures**: https://anonymous.4open.science/r/Rebuttal-Figures-for-ICML-Submission-19687-78CF/README.md
>
> **W1:**
>
> > A more convincing analysis would compare the extracted CARD vector with the latent representation shift induced by visual inputs.
>
> We provide three new pieces of evidence that CARD captures meaningful visual information:
>
> **(1) Semantic Clustering:** We extracted CARD vectors for 4 MSCOCO categories (airplane, elephant, pizza, train; 50 images each). t-SNE shows clear clustering: airplane images produce similar CARD vectors, elephant images produce different ones, etc. This directly proves CARD encodes visual content, not noise. (See Fig. R1 in the link.)
>
> **(2) Residual Direction Alignment [Fig R2, center panel]:** For each CHAIR image, we extracted CARD with and without the image. We then asked: Does the image-conditioned CARD reliably capture the change caused by adding that image? For each sample, we measured the cosine similarity between the image-conditioned CARD and the shift it induced ($v_{\text{CARD}}^{\text{img}} - v_{\text{CARD}}^{\text{text}}$). Result: 0.548 ± 0.011 across 40,137 samples. This means that for every image, CARD faithfully encodes the visual shift introduced by that image, with remarkable consistency (std = 0.011) across all samples. (See Fig. R2 center in the link.)
>
> **(3) Rotation Angle:** We asked: how much does adding an image change CARD? The angle between text-only and image-conditioned CARD is 66.4° ± 1.5° on LLaVA-1.5-7B. The tight spread (std = 1.5°) means every image rotates CARD by almost the same amount, not noise. (See Fig. R2 right in the link.)
>
> These complement the existing analysis in Appendix A.4, summarized for convenience:
>
> - Fig 5a: t-SNE overlay showing each image's CARD systematically rotates away from its text-only counterpart: paired lines connect the same sample, revealing a consistent directional shift.
> - Fig 6a: This rotation is ~40° with tight distribution on Idefics2, confirming it is not random.
> - Fig 6b: The image-conditioned CARD is closer to the actual steering direction than the text-only CARD (alignment gain = 0.239), validating that CARD captures the signal the Beta Gate relies on.
>
> ---
>
> **W2:**
>
> > The paper does not verify if this configuration is near-optimal for POPE and MME.
>
> We ran a grid search of (α_max, k) on POPE for LLaVA-1.5:
>
> ```
> POPE (%)    | α=10  15   20*  25   30  |  α=10  15   20*  25   30
>             |        Accuracy          |          F1
> k=3         | 86.1 86.3 86.4 86.4 86.6 | 85.2 85.4 85.6 85.7 86.1
> k=5*        | 86.2 86.4 86.5 86.6 86.7 | 85.2 85.5 85.7 85.9 86.1
> k=8         | 86.3 86.4 86.5 86.7 86.7 | 85.4 85.5 85.7 86.2 86.1
> Baseline:   |        85.3%             |        84.7%
> *CHAIR-optimal. All configs > baseline. Acc spread: 0.6pp.
> ```
>
> All 15 configs beat the baseline. The CHAIR-tuned setting corresponds to (α_max=20, k=5) and achieves 86.5%, which is within 0.2pp of best. Hence, model-specific tuning on one dataset transfers well.
>
> The purpose of the MME experiment was to show that our method both reduces hallucinations and preserves general abilities (recognizing colors, counting numbers) on a separate target set, so no further hyperparameter tuning was needed.
>
> ---
>
> **W3:**
>
> > Other types of multimodal errors are not thoroughly evaluated.
>
> We agree that extending to attribute/relational hallucination is valuable. MME (Table 3) includes reasoning and attribute subtasks where RUDDER preserves or improves scores. We will discuss this.
>
> ---
>
> **W4:**
>
> > It remains unclear if the proposed method generalizes to more recent architectures.
>
> Our evaluation includes Qwen2.5-VL-7B (Table 5), which is a recent architecture. RUDDER achieves the lowest CHAIR_I (7.0) while improving POPE F1 (88.1 vs 87.8). We will add Qwen3-VL to the final version.
>
> ---
>
> **W5:**
>
> > The paper lacks detailed analysis or visualization of how the gating mechanism behaves during generation.
>
> We recorded per-token gate values during generation [Fig R3, in the link]. Example from a skiing image — both the gate g_t and cosine similarity cos(h_t, v_CARD) track visual evidence strength:
>
> ```
> Token    |  g_t | cos  | Evidence
> skiing   | .836 | .287 | ★ visible in image
> downhill | .816 | .264 | ★ visible
> flags    | .809 | .258 | ★ visible
> poles    | .789 | .236 | ★ visible
> appears  | .672 | .130 | ✗ hedging word
> control  | .664 | .122 | ✗ abstract concept
> high     | .656 | .115 | ✗ not observable
> man      | .652 | .112 | △ small/blurry in image
> ```
>
> Directly visible objects get high values; abstract/inferred concepts and hard-to-see objects get low values. The gate tracks visual grounding, not just syntactic category.
>
> ---
>
> **W6:**
>
> > Including additional hallucination mitigation baselines in the qualitative comparisons would provide a clearer illustration.
>
> We will add VISTA/VCD comparisons in the final version.

---

> > ### Author Rebuttal · Reviewer_jMxp · 2026-04-02
> >
> > Thank you for the detailed rebuttal and additional experiments. My concerns are largely addressed, and I will update my score to 4 (Weak Accept). I encourage the authors to incorporate these additional results into the final version to further strengthen the paper.

---

> > > ### Author Response · Authors · 2026-04-02
> > >
> > > Thank you for the updated assessment. We are glad the new experiments addressed your concerns. We will incorporate all additional results (semantic clustering, CARD alignment analysis, gate visualizations, and POPE sensitivity) into the revised paper, along with the writing improvements discussed.

---

### Official Review · Reviewer_TKab · 2026-04-04

**Soundness:** 3
**Presentation:** 3
**Significance:** 3
**Originality:** 3
**Overall Recommendation:** 4
**Confidence:** 4

**Summary:**

This paper proposes RUDDER, a training‑free method for reducing object hallucinations in large vision‑language models (LVLMs). The method requires no extra forward passes, adds negligible latency (~96% of vanilla throughput), and is evaluated on CHAIR, POPE, and MME across several LVLMs (LLaVA‑1.5, Idefics2, InstructBLIP, Qwen2.5‑VL). Results show competitive or better hallucination reduction compared to prior training‑free methods like VISTA, VCD, and DeGF.

**Compliance With Llm Reviewing Policy:**

Affirmed.

**Final Justification:**

In the second round of rubuttal, most of the concerns have been well addressed. I am keeping my original score.

**Key Questions For Authors:**

please refer to the weakness part.

**Limitations:**

yes

**Strengths And Weaknesses:**

Strengths:
1. Technical intuition – The idea of extracting a visual direction from residual updates during the mandatory prefill is clever and well motivated by internal dynamics analysis (Fig. 7). The Beta Gate as a trust mechanism (rather than error detector) is a nice conceptual twist.
2. Empirical scope – Evaluation covers multiple LVLMs (including 13B and Qwen2.5‑VL), three decoding strategies, and both sentence‑level (CHAIR) and binary‑choice (POPE) tasks. Ablation studies are also provided.
3. Clarity of presentation – The paper is generally well structured. Figures and tables are readable, and the algorithm pseudocode is also provided.

Weaknesses:
1. Limited methodological novelty: The core components – steering using a precomputed direction, adaptive gating based on similarity, and adding a vector to residual streams – are already present in prior work (e.g., VISTA, activation steering, contrastive decoding).
2. Marginal empirical gains: On CHAIR (Table 1), RUDDER‑Beta is often statistically comparable to VISTA (e.g., 33.1 vs 33.9 on LLaVA‑1.5 with nucleus sampling). On POPE (Table 2), differences are very small (e.g., 86.53 vs 86.21 accuracy). The MME scores (Table 3) sometimes drop slightly below vanilla. The paper claims “consistently superior”, but the improvements are modest and not always significant.

---

### Decision · Program_Chairs · 2026-04-30

**Decision:**

Accept (regular)

**Comment:**

This paper proposes RUDDER, a training-free inference-time framework that extracts a visual evidence direction (CARD) from prefill residual updates and injects it during decoding via an adaptive Beta Gate to reduce object hallucination in LVLMs with minimal computational overhead. Reviewers initially raised concerns around limited novelty, insufficient empirical validation of CARD as a meaningful visual direction, unclear layer selection rationale, and the counterintuitive gating behavior when visual alignment is low, though one reviewer (jMxp) upgraded from fair to weak accept after the rebuttal, which provided t-SNE clustering, alignment analysis, token-level gate visualizations, and cross-task hyperparameter sensitivity results that adequately addressed most concerns. For the final version, we recommend the authors incorporate these rebuttal experiments into the main paper, strengthen the preliminary section with clearer definitions of the residual stream and prefill vs. decode distinction, and move the efficiency table to a more prominent position.